# Anti-Inflammatory Properties of Diet: Role in Healthy Aging

**DOI:** 10.3390/biomedicines9080922

**Published:** 2021-07-30

**Authors:** Kristine Stromsnes, Angela G. Correas, Jenny Lehmann, Juan Gambini, Gloria Olaso-Gonzalez

**Affiliations:** 1Freshage Research Group, Department of Physiology, Faculty of Medicine, Insitute of Health Research-INCLIVA, University of Valencia and CIBERFES, Avda. Blasco Ibañez, 15, 46010 Valencia, Spain; krisbaks@alumni.uv.es (K.S.); angela.garco.96@gmail.com (A.G.C.); gloria.olaso@uv.es (G.O.-G.); 2Department of Molecular Toxicology, German Institute of Human Nutrition, Potsdam-Rehbrücke, Arthur-Scheunert-Allee 114-116, 14558 Nuthetal, Germany; jenny.lehmann@uni-potsdam.de; 3Institute of Nutritional Science, University of Potsdam, 14558 Nuthetal, Germany

**Keywords:** anti-inflammatory, diet, polyphenol, inflammation, aging, inflammaging, nutraceutics, nutrition, healthy aging

## Abstract

Inflammation is a physiological process involved in the defenses of the body and the repair of tissues. It is acutely activated by infections, trauma, toxins, or allergic reactions. However, if it becomes chronic, inflammation can end up stimulating the development of diseases such as cardiovascular disease, autoimmune disease, neurological disease, or cancer. Additionally, during aging, inflammation becomes increasingly more chronic. Furthermore, we found that certain foods, such as saturated fats, have pro-inflammatory activity. Taking this into account, in this review we have discussed different diets with possible anti-inflammatory activity, the commonly ingested components of each diet and their active compounds. In addition, we have proposed some dietary guidelines, as well as a list of compounds present in foods with anti-inflammatory activity, outlining how to combine them to achieve optimal anti-inflammatory effects. Therefore, we can conclude that the compounds in our diet with anti-inflammatory activity could help alleviate the inflammatory processes derived from diseases and unhealthy diets, and thereby promote healthy aging.

## 1. Aging and Inflammation: Inflammaging

Aging is a physiological process characterized by deleterious changes occurring in body systems with advancing age, affecting their functionality and producing an increased vulnerability to death. It is a complex multicomponent phenomena attempted to be explained by different complementary theories such as the free radical theory, the endocrine and the immunological theories of aging [1]. There seems to be an agreement that the heterogeneity of the aging phenotype is the result of both genetic and environmental factors. In 2013, nine hallmarks of aging were defined: genomic instability, telomere attrition, epigenetic alterations, loss of proteostasis, deregulated nutrient-sensing, mitochondrial dysfunction, cellular senescence, stem cell exhaustion, and altered intercellular communication [2]. In addition, inflammation seems to play a relevant role in aging as well.

Inflammation is a physiological process involved in the defenses of our body and the repair of tissues. It is acutely activated by infections, trauma, toxins, or allergic reactions. However, when it becomes chronic, it can end up stimulating the development of diseases such as cardiovascular disease, autoimmune disease, neurological disease, or cancer. It can be activated in vascular connective tissues or by the immune system itself. Tissue activation is characterized by the mobilization of phospholipids present in plasma membranes induced by phospholipase A2 (PLA2), which specifically cuts phospholipids, transforming them into arachidonic acid (AA). This can follow two routes: that of lipoxygenases (LOX) that will form leukotrienes, and that of cyclooxygenases (COX) that can form prostaglandins (PGs) and thromboxanes (TXs). Products formed from AA can have pro- or anti-inflammatory activity, thereby having the ability to regulate inflammation. Another pathway involved in inflammation is mediated by NF-κB (nuclear factor kappa b); this protein is a nuclear factor that—among its functions—is to activate the gene expression of inflammation factors such as TNF-α (necrosis factor tumor alpha), IL (interleukins), COX (cyclooxygenases), iNOS (inducible nitric oxide synthase) or adhesion factors such as ICAM (Intercellular Adhesion Molecule 1) or VCAM (Vascular Adhesion Molecule).

Inflammaging is a term which refers to the chronic inflammatory state associated with aging in mammals [3]. According to the defined hallmarks of aging, inflammaging can be considered an alteration in intercellular communication which led to a pro-inflammatory phenotype [2]. Different processes including the propensity of senescent cells to secrete proinflammatory cytokines, the enhanced activation of the NF-κB transcription factor, and the occurrence of a defective autophagy response produce an enhanced activation of the NLRP3 inflammasome and other pro-inflammatory pathways which leads to increased production of IL-1β, tumor necrosis factor and interferons [3,4].

The influence of dietary patterns on systemic markers of inflammation like IL-6, TNF-α and C-reactive protein (CRP) have been determined in human studies [5]. Different chemical compounds present in diet can modulate proinflammatory pathways [6]. The glycemic index (GI) and load (GL), fiber, fatty acid composition, magnesium, carotenoids, and flavonoids are only some of them.

## 2. Diet and Inflammation

To understand the possible anti-inflammatory properties of diet on health and aging, it is necessary to perform studies that analyze the effects of dietary patterns on a specific population. Analyzing the relationship between a diet with the pathophysiology of different diseases is complicated. The focus of current research lays not only on the role of single nutrients and foods, but also on the compilation and the entire process from food production to food consumption [7]. The general idea is that food items show either synergistic or antagonistic effects when consumed in combination.

The dietary inflammation index (DII) was developed to enable the analysis the pattern of a whole diet on inflammation markers circulating in blood plasma [8]. According to the DII, some diets could be classified as pro-inflammatory or anti-inflammatory. Although it is true that in industrialized countries, we find individuals with more or less healthy diets, the main recognized pro and anti-inflammatory diets whose characteristics are discussed below (Figure 1).

In this paper, we have focused on reviewing the most significant articles published in the last five years related to the characteristics of well-known pro-inflammatory and anti-inflammatory diets, analyzing which of its components possess inflammatory modulators and the mechanism through which they exert said function.

Finally, we propose an anti-inflammatory dietary pattern to reduce the inflammation associated with aging, including specific recommendations about the consumption of foods rich in anti-inflammatory compounds.

### 2.1. The Western Diet: An Example of Proinflammatory Dietary Pattern

Since the industrial revolution 150 years ago, the population of industrialized countries has gained unlimited access to a high caloric diet. This resulted in excess food consumption, along with technological advances, which facilitated a sedentary lifestyle, which ultimately led to obesity and the development of non-communicable diseases, e.g., metabolic syndrome. The overall composition of this dietary pattern is a high intake of refined grains and sugars, red and processed meat, eggs, high fat dairy products, artificially sweetened drinks, high consumption of salt and a low intake of fruits, vegetables, whole grains, fish, nuts, and legumes, which contribute to weight gain and the proliferation of visceral adipose tissue [9]. Adipose tissue is an endocrine organ which significantly contributes to inflammatory processes though releasing proinflammatory factors including the adipokine leptin [10], as well as TNF-α and IL-6 [11,12]. Furthermore, the modern Western diet is rich in advanced glycation end products (AGEs) [13]. The correlation between advanced glycation end products in the progression of age-associated diseases has widely been reported [14] along with their pro-inflammatory properties [15]. Not only the common ingredients of Western diet, but also the way of cooking them, i.e., fried, contribute to the increase of levels of AGEs. For instance, in a pilot study performed in patients undergoing peritoneal dialysis, ten of the participants were told to follow their normal diet, whereas another group of ten patients were told to prepare their food by boiling or steaming, and to avoid fried entrees. After only a one-month period, the microbiota was enriched in bacteria related to better health, and AGE levels were reduced in the intervention group [16].

Abdominal and visceral obesity has tripled in the U.S. population in the last 40 years and estimations predict a worldwide obesity rate of 1.12 billion by 2030 [17]. Data also shows that only 5–10% of cancer is caused by genetics, whereas 90–95% arise due to environmental factors and lifestyle, e.g., the diet we regularly consume [18]. Due to a significant increase in these diseases in the past decades, they represent a rising burden for the public health system with the dimensions of a global pandemic and rank at the top of the leading causes of mortality in the western world [19]. Although the life expectancy in most of the industrial nations are similar, providing individuals with the resources and information to facilitate healthy aging and thereby avoiding the burden of these common diseases is of utmost importance.

### 2.2. Anti-Inflammatory Diets

#### 2.2.1. Mediterranean Diet

The blue zones around the globe have provided some answers to research the potential sources for healthy aging. Five different locations worldwide have been described in which the concentration of centenarians is higher than usual. Further investigation found that the populations in these specific communities share nine characteristics, of which three are connected to dietary habits, such as the 80% rule, the diet being mostly plant based, and moderate wine consumption [20]. Sardinia (Italy) and Icaria (Greece) represent the blue zones in the Mediterranean area and make the traditional local diets the most researched dietary pattern of the world. The Mediterranean Diet (MD) was first introduced from Ancel Keys through his discoveries of its health benefits on cardiovascular diseases in the Seven Countries Study, which started in 1956 and was the world’s first epidemiological study from multiple countries [21].

The MD is defined through its ingredients, methods of conservation and culinary processes, which are passed on through generations before the globalization of the food system. The MD is described in civilizations that surround the Mediterranean Sea because they share common foods and cooking methods that comprise the dietary pattern. Differences in culture, religion and economy explain some variations, but the commonly regularly consumed ingredients include, olive oil, legumes and nuts, unrefined cereals, fruits vegetables, moderate consumption of fish, seafood and dairy products, low consumption of alcohol and a limited use of red and processed meat.

The long-term health benefits of MD have been widely studied thanks to the PREDIMED study, a multicenter, randomized, primary prevention trial [22]. Olive oil is especially emphasized due to its ability to reduce radical oxidative species (ROS) and prevention of inflammation related diseases due to its phenolic compounds [23]. In a multicenter trial performed in Spain, 7447 participants between the ages of 55 and 80 with high cardiovascular disease (CVD) risk were assigned one of three diets: MD with extra-virgin olive oil, MD with nuts, or a control diet of reduced fat intake. After a 5-year period, both groups following a MD showed lower risk of cardiovascular events by 1.7–2.1%, whereas the low-fat diet control group showed no significant improvements. Authors attributed the protective role shown by MD to their antioxidants and anti-inflammatory components [24]. There is a similar study where 165 high CVD risk participants were assigned the same three diets. After 3 and 5 years, both groups following the MD had significant reductions of the inflammatory markers CRP, IL-6, TNFα, and monocyte chemoattractant protein 1 (MCP-1) in plasma by ≥16% [25]. Lopez-Moreno et al. developed a new randomized cross-over study where 20 healthy older adults were assigned three diets: a MD enriched with virgin olive oil, a diet rich in saturated fatty acids, and a low-fat high-carbohydrate diet enriched with ω-3 polyunsaturated acids. After a 3 week period, the postprandial intestinal absorption of lipopolysaccharides (LPS) in the low-fat diet increased inflammatory response compared with the MD [26].

Different studies have shown a direct relationship between arachidonic acid (AA) levels and chronic inflammation, a condition that occurs in obesity and liver disease [27]. Tutino et al. studied the effects of a MD or control diet, alone and in combination with physical activity programs, on the arachidonic/eicosapentaenoic acid ratio (AA/EPA), a common biomarker used to evaluate inflammation, in 146 patients with nonalcoholic fatty liver disease. Significant reduction in AA/EPA ratio were observed both at 45 and 90 days in the group assigned a MD combined with an aerobic activity program, while improvements were observed only after 90 days in the group assigned a control diet and the same activity program [28].

The apparent anti-inflammatory action of MD has been validated in more clinical trials with subjects with different characteristics. In a randomized controlled trial, 99 volunteers with osteoporosis were randomly assigned into groups of intervention (*n* = 50) and control (*n* = 49). The intervention group were asked to follow a MD, and after the 16 week trial they showed significantly decreased levels of the pro-inflammatory cytokine IL-1α [29]. Another small scale randomized clinical trial was performed in 34 participants of nonrestricted MD and 31 participants following a low-fat diet. Those following the MD had reduced subcutaneous fat and waist circumference after the 6 month period, which discounts the misconception that diets high in fats lead to weight gain [30].

The microbiota can modulate changes in aging related to innate immunity, sarcopenia, and cognitive function, which are essential components of the frailty syndrome through nutrition we can modify the microbiota to maintain intestinal health by stimulating beneficial bacteria [31]. In a parallel 8-week randomized controlled trial, 82 overweight and obese subjects without underlying diseases were separated into two groups. Forty-three subjects were set to follow a MD tailored to their habitual energy intakes, and 39 subjects maintained their regular diets. After the 8 week period, a significant increase in gut microbial gene richness and lower levels of serum hs-CRP was found in the diet intervention group, whereas no changes were found in the control group [32]. Changes in the microbiota with apparent anti-inflammatory effects have been also found in another recent clinical trial, where the microbiota of 612 nonfrail or pre-frail subjects across five European countries (UK, France, Netherlands, Italy, and Poland) was profiled before being administered a tailored MD tail. After a 12-month period, the enrichment of the microbiome taxa was associated with lower frailty markers, CRP, and interleukin-17 [33]. In another small-scale trial, 65 participants with diagnosed coronary heart disease were randomly assigned a MD or a low-fat diet intervention for 6 months. In the 56 subjects that completed the trial, the dietary inflammatory index was significantly reduced, through reductions in high sensitivity interleukin-6 and triglycerides levels [34]. Similarly, in a study with type 2 diabetes patients, the MD intervention group showed improved endothelial function and reduced CRP and ICAM-1 fasting blood levels after the 3 month trial [35].

The anti-inflammatory effects of MD, according to the referred results found in clinical trials performed in the last 5 years, are unquestionable. However, even though the health benefits of the MD have been known around the world for many years, adherence to this diet has been difficult for Western communities. Disadvantages are the limited availability and variety of ingredients outside of the Mediterranean area, higher costs, and a lack of familiar tastes [36]. To provide a healthy and easy-to-follow diet for the populations of different locations multiple diet pattern have been constructed. Other researched diet designs, such as the Nordic diet, traditional Asian diets such as the Japanese and Tibetan ones, have been put to focus and are also associated with a reduced risk of cardiovascular disease and other inflammatory-derived diseases [37].

#### 2.2.2. Nordic Diet (ND)

Similar to the MD, the Nordic Diet (ND) focuses on the consumption of fruits and vegetables, which are replaced with local grown ingredients, such as berries, apples, pears, carrots, potatoes, cabbages and whole grain products, local fish products, the restriction of saturated fats as well as red and processed meats. The olive oil as the main source for unsaturated fats of the Mediterranean diet is replaced with canola oil [38].

The anti-inflammatory characteristics of ND has also been studied. In 2019, Tuomainen et al. set out to investigate the association between whole grain derived compounds and metabolic health from blood samples of 163 participants of the SYSDIET study [39]. They found clear associations between the healthy Nordic diet and the inflammation biomarkers IL-1 receptor agonist and CRP. Furthermore, Roager et al. showed how a whole grain diet significantly decreases IL-6 and CRP in 60 Danish adults with metabolic syndrome risk in an 8-week randomized cross-over trial [40].

In another substudy of the SYSDIET study, 88 patients with metabolic syndrome were assigned a ND or control diet consisting of low-fiber cereal products, milk fat, and restricted amounts of fish and berries. After a 18–24 week period, the expression level of the gene TNF-receptor superfamily member 1A was downregulated, although the NF-κB subunit, RELA proto-oncogene, was up-regulated in the intervention group [41]. Leder et al. evaluated whether the ND can modify the expression of inflammation-related genes in peripheral blood mononuclear cells by performing a 2 h oral glucose tolerance test in 89 participants from the SYSDIET study from three Nordic centers. They found significant downregulation of toll-like receptor 4, IL-18, and CD36 and upregulated expression of peroxisome proliferator-activated receptor delta (PPAR-δ) in the Nordic diet group compared to the control group [42].

The health benefits of ND recently gained interest in the scientific community, and therefore, the number of clinical trials performed is low compared to those performed with MD. However, the results found are very promising and point to an anti-inflammatory role of different components of this diet.

#### 2.2.3. Asian Diet

##### Japanese Traditional Diet: Washoku

Japan is known for its exceptionally healthy diet and extended longevity. In fact, Okinawa, the islands at the southern of Japan, is one of the seven blue zones and was once called the land of immortals. The traditional Japanese diet (Washoku) is characterized by high consumption of many types of fish, rich in EPA and docosahezaenoic acid (DHA), and soybean products and low consumption of animal fat and meat. The use of umami to enhance their recipes palatability avoids the consumption of refined sugar and salt. Steaming, boiling, and stewing are the main cooking methods in this diet, and therefore, their dishes are rich in water and low in fat, and subsequently have low-calorie density. The portion sizes usually are small, and seasonally available vegetables are typically used [43].

The effects of Japanese diet on inflammation have been studies through different clinical trials. An example is the work developed by Coe et al. who studied the blood samples of 382 middle aged and elderly adults in Japan to determine the effects of Japanese diet customs, with special emphasis on tea, vegetable, and seafood consumption. They found that the consumption of a Japanese diet was associated with significantly lower IL-6 and CRP levels [44]. Additionally, another study also found that subjects following a Japanese diet had decreased HbA1c levels compared to a control group already after 2 weeks. VFA levels, which have been shown to be associated with inflammation and CRP levels, were also decreased, possibly via the suppression of glucose-dependent insulinotropic polypeptide [45,46,47].

Despite the benefits of the traditional Japanese diet, like the forementioned diets, this also has become Westernized over the past decades and consequent disease rates have risen. A small-scale randomized controlled trial in 21 subjects was performed to evaluate the effects of the modern and traditional Japanese diets. The researchers found that the 1975 Japanese diet decreases bacteria associated with lifestyle diseases in 11 subjects compared to the 10 subjects following a modern Japanese diet for 28 days. Four genera, including *Parabacteroides* were changed in the 1975 diet group, which are known to promote inflammatory bowel disease. HbA1c, which is correlated with inflammation, was also decreased on those following the 1975 diet [48,49].

##### Chinese Traditional Diet: Jiangnan Diet

Finally, although it has not been studied deeply yet, the characteristics of a traditional Chinese based diet, the Jiangnan diet, used in some regions of China nowadays, are worth mentioning.

The dietary patterns and nutrition composition of the Chinese diet have changed considerably from the 80s, becoming increasingly more Westernized [50]. The consumption of refined grains, red meat, processed meat, and sugar-sweetened beverages has increased, and, furthermore, frying is more used for cooking than the traditional boiling or steaming. Consequently, the prevalence of chronic noncommunicable diseases (NCDs), including obesity, diabetes, CVD, and cancers, have increased [51,52]. However, there seems to be differences between the Northern regions and a region in the South of China, where obesity and metabolic syndrome prevalence are lower in the latter. This difference has been associated to the diverse dietary pattern between the two regions [50]. In the South of China, reminiscences from traditional Chinese food patterns remain and steaming or boiling are the preferred cooking procedures. Furthermore, in that zone there is a higher consumption of vegetables and fruits in season, fresh fish, legumes, moderate consumption of whole-grain rice, plant oils (mainly rapeseed oil), and red meat, and relatively low consumption of salt. The consumption of soy-derived products is also common in this region.

The anti-inflammatory properties of Jiangnan Diet have not yet been studied, but according to their ingredients and methods of cooking, as well as the epidemiologic data, it is very promising. Nevertheless, more investigation to confirm its beneficial effects on health is necessary.

## 3. The Anti-Inflammatory Compounds Present in the Diet: Classification and Mechanism of Action

### 3.1. Classification

The anti-inflammatory properties of the diets discussed above are due to the presence of compounds in the most commonly ingested foods of each region. These properties are mainly attributed to phenolic structures present in fruits and vegetables, and to a lesser extent in cereals, legumes, and fish. Table 1 shows different compounds with anti-inflammatory activity and the foods with the highest proportions.

To explain the mechanism of action through which the selected anti-inflammatory compounds run their function, we have classified them according to their chemical structure.

### 3.2. Mechanism of Action

#### 3.2.1. ω-3 Polyunsaturated Fatty Acids (ω-3 LC-PUFAs)

Polyunsaturated fatty acids (PUFAs) of n-3 series, also known as omega-3 (ω-3) fatty acids, exert important biological actions in blood coagulation and the resolution of inflammatory processes [58]. The anti-inflammatory effects of ω-3 PUFAs are due to multiple interlinked mechanisms, comprising alterations in phospholipid fatty acid composition of the cell membrane lipid rafts, inhibition of NF-κB signaling which triggers the activation of proinflammatory mediators, activation of the anti-inflammatory transcription factor PPAR-γ, and the binding to the ω-3 PUFA membrane receptor GPR120 which represses macrophage-mediated tissue inflammation [59]. Figure 2 summarizes the main anti-inflammatory actions of ω-3 PUFA.

Linoleic acid (C18:2, ω-6) and α-linolenic acid (C18:3, ω-3) are essential fatty acids for humans, and are necessary for omega-6 and omega-3 fatty acid synthesis, respectively, such as eicosapentaenoic acid (EPA; C20:5, ω-3), docosahexaenoic acid (DHA, C22:6, ω-3), arachidonic acid (AA; C20:4, ω-6) and docosapentaenoic acid (DPA; C22:5, ω-6), which are precursors of eicosanoids (prostanoids and leukotrienes).

Eicosanoids like AA (ω-6) are strong pro-inflammatory mediators. However, EPA (ω-3) and DHA (ω-3) generate mediators with anti-inflammatory properties or weaker inflammatory actions, even when they bind to the same receptor [60].

AA can be metabolized by cyclooxygenases (COX-1 and COX-2) enzymes giving rise to 2-series prostanoids, including prostaglandins (PGE2, PGI2, PGD2, PGF2α) and thromboxanes (TXA2, TXB2). Additionally, it can be metabolized by lipoxygenase (LOX), generating 4-series leukotrienes (LTA4, LTB4, LTC4, LTD4), many of which have strong pro-inflammatory effects, including vasoconstriction, platelet activation and aggregation, and neutrophil and macrophage activation [61,62,63,64].

Whereas 3-series prostanoids (PGE3, PGI3, TXA3) and 5-series leukotrienes (LTA5, LTB5, LTC5, LTD5) derived from EPA and DHA metabolism exert mostly anti-inflammatory actions with antithrombotic and vasodilative effects, thus antagonizing the effects of arachidonic acid-derived eicosanoids [58,60,65]. Moreover, arachidonic acid-derived eicosanoid production is counteracted by EPA, as it is a competitive inhibitor for the enzymes involved in the signal pathway [66].

Other bioactive lipid mediators generated from EPA and DHA metabolism through COX and LOX enzyme pathways are called resolvins (Rvs), protectins (PDs) and maresins (MaRs). They have anti-inflammatory and inflammation resolving effects, as well as immunoregulatory actions on T and B cells to promote the resolution of inflammation. E-series resolvins are produced from EPA, and D-series resolvins, protectins and maresins are produced from DHA.

Resolvin E1 (RvE1), resolvin D1 (RvD1) and protectin D1 inhibit diapedesis or transendothelial migration of neutrophils, preventing their infiltration into the inflamed tissue. Furthermore, they inhibit pro-inflammatory cytokines production, including IL-1β and TNF-α, lower the production of ROS and promote macrophage efferocytosis of apoptotic neutrophils. Maresin 1 (MaR1) additionally stimulates apoptotic neutrophil clearance [67,68].

ω-3 PUFAs also have an impact on lipid rafts formation that act as signaling platforms and protein recruitment sites. Lipid rafts are rich in cholesterol, for which ω-3 PUFAs have low affinity. Thus, instead of direct incorporation of dietary PUFAs into the lipid rafts, they incorporate into other regions of cell membranes from where they seem to influence lipid rafts function and formation, hampering protein recruitment and cell signalling [69].

In addition to the abovementioned biochemical competition, another mechanism of the ω-3 involves the NF-κB, a crucial transcription factor in inflammatory responses that upregulates the expression of a variety of proteins involved in the inflammatory processes (Figure 2), including chemokines and cytokines (such as TNF-α, IL-1β, IL-6, IL-8, IL-12p40, MCP-1, MIP-1), adhesion molecules (ICAM-1, VCAM-1, selectins), enzymes (COX-2, iNOS) and platelet activating factor. The NF-κB transcription factor is localized in the cytosol associated with inhibitor proteins IκBs as an inactivated trimer. The phosphorylation of the inhibitory subunit IκB by IκB Kinases (IKKs) triggers its dissociation and translocation of the remaining active dimeric NF-κB to the nucleus, activating pro-inflammatory target genes [70,71].

In response to certain stimulus, the downstream signaling pathway of some Pattern Recognition Receptors (PRRs), including the toll-like receptor (TLR) family, activates transforming growth factor-β-activated kinase 1 (TAK1), a member of the MAP3K family essential for TLR-mediated activation of NF-κB. TAK1 activation requires assembling with TAK-1 binding proteins (TAB1, TAB2 and TAB3). The resulting TAK1-TABs complex phosphorylates the IκB Kinase β (IKKβ), thus leading to NF-κB activation [70,71].

The NF-κB signaling pathway can be inhibited by ω-3 LC-PUFAs through interacting with G-protein coupled receptor 120 (GPR120), also known as FFA4, whose activation induces the recruitment of β-arrestin-2 and the internalization of the receptor complex. In turn, β-arrestin-2 recruits TAK1/2 binding protein (TAB1/2), an element of pro-inflammatory pathways such as TLR2/4 and TNF-α pathways, which interferes and disassembles their cascades, resulting in reduced inflammation [66,72]. Following GPR120 ω-3 LC-PUFA-induced activation, β-arrestin-2 can also bind to nucleotide-binding oligomerization domain-like receptor containing pyrin domain 3 (NLRP3) protein, disrupting the inflammasome structure and inhibiting NLRP3 inflammasome activation [72,73]. An additional mechanism by which NF-κB signaling is inhibited involves PPAR-γ, a transcription factor with anti-inflammatory effects that can be activated by EPA and DHA. PPAR-γ physically interferes with NF-κB translocation to the nucleus, thus inhibiting its signaling and reducing pro-inflammatory cytokine expression [74,75].

#### 3.2.2. Phenolic Compounds and Polyphenols

Phenolic compounds are a large heterogeneous group of molecules widely distributed in nature. Polyphenols and phenolic compounds are an important group of secondary metabolites, found in plants, including fruits and vegetables, tea, coffee, chocolate, herbs and spices, whole grains, edible mushrooms and fungal fruiting bodies, and red wine. Phenolic compounds are classified according to their chemical structure, based on the number of aromatic rings with attached hydroxyl groups (Figure 3).

Numerous in vitro and in vivo studies highlight their antioxidative and anti-inflammatory properties, flavonoids being the most studied polyphenols [76,77,78]. They exert these properties through different mechanisms, including antioxidant activity restoration, inhibition of pro-inflammatory enzymes, and modulation of mediators and transcription factors involved in inflammatory processes [79]. Although the precise mechanisms of action of phenolic compounds have not been fully dilucidated, a correlation between the high intake of phenols and polyphenol-rich food and a downregulation of inflammatory processes have been found [80]. Therefore, a high enough daily intake of this group of compounds could ameliorate inflammaging and inflammation associated with chronic diseases.

Phenolic compounds exert their anti-inflammatory properties through multiple ways, such as inhibiting the activity, gene expression or synthesis of pro-inflammatory mediators. They act on COX-2, iNOS, and eicosanoids, thereby inhibiting the immune cell activation, modulating transcription factors, like NF-KB or Nrf-2, which result in anti-inflammatory and antioxidant effects [80]. In addition to a reduction of pro-inflammatory markers, such as IL-1β, 1L-6, TNF-α, phenolic compounds also lower LDL oxidation, leading to a decreased vascular inflammation, risk of platelet aggregation, and a reduction in oxidative stress and nitric oxide (NO) effects [80]. Figure 4 summarizes the anti-inflammatory properties of these compounds.

As previously mentioned, the arachidonic acid released from the cell membrane phospholipids by PLA2 is metabolized by COX and LOX enzymes to prostaglandins, thromboxanes and leukotrienes. Flavonoids can inhibit the enzymatic activity of PLA2, COX and LOX enzymes as well as inhibit protein expression of COX and LOX, quercetin being the first flavonoid described to inhibit PLA2 in human polymorphonuclear leukocytes. Flavonoids can also interfere with signaling pathways, including protein kinase C, NF-KB and tyrosine kinase pathways, leading to the suppression of COX gene expression, though the inhibiting mechanisms of the enzymatic activity have not been fully elucidated [81,82]. Flavones like apigenin and luteolin, flavonols, isoflavones like genistein, flavanols like catechin, among other examples gathered in Table 2, have also demonstrated the capacity to inhibit COX and/or LOX enzymes both in in vitro and in vivo models [82]. Furthermore, NF-κB activation, iNOS expression, NO production, aldosterone signaling, and gene expression induced by aldosterone and cytokines production (TNF-α, IL-1β) can also be inhibited by flavonols like kaempferol [80].

The anti-inflammatory properties of phenolic and polyphenolic compounds can also be attributed to their antioxidant activities. Flavonoids exert antioxidant activities by scavenging ROS generated by neutrophils and macrophages and inhibiting the ROS-producing enzymes NADPH oxidase, xanthine oxidase and myeloperoxidase [82]. Additionally, flavonoids can activate Nrf2 signaling pathway [81], thereby exerting an anti-inflammatory role. Nrf2 binds to antioxidant responsive elements (AREs), thus activating expression of genes such as HO-1, which directly inhibits pro-inflammatory cytokines and activates anti-inflammatory cytokines. HO-1 also catalyzes the heme into free iron, biliverdin and carbon monoxide (CO); CO inhibits NF-KB pathway. Thus, Nrf2 leads to a decreased expression of pro-inflamamtory mediators and enzymes (COX, LOX, iNOS). Nrf2 inhibits cytokines and chemokines production, including TNF-α, IL-1β, IL-6, IL-17, MPC1, MIP2, as well as inhibits gene expression of CAMs induced by these cytokines [83].

#### 3.2.3. Terpenes and Terpenoids

Terpenes and terpenoids are a large group of compounds mainly derived from plant secondary metabolism that have demonstrated anti-inflammatory properties, both in vitro and in vivo, through regulating levels of pro-inflammatory mediators and transcription factors, interfering with signaling pathways and reducing oxidative stress. The structure of terpenes is based on the hydrocarbon 2-methyl-1,3-butadiene or isoprene (C5H8). According to the number of isoprene molecules that constitute them, terpenes are classified in monoterpenes (C10), sesquiterpenes (C15), diterpenes (C20), sesterterpenes (C25), triterpenes (C30), sesquarterpenes (C35), tetraterpenes (C40) and polyterpenes. Terpenoids are terpenes derived compounds containing additional functional groups [84]. Examples of terpenes with anti-inflammatory actions include d-limonene, acanthoic acid, terpinolene, linalool, α-pinene, α-terpineol, α-phellandrene, and γ-terpinene [84,85,86].

As briefly explained in previous sections, inflammatory responses are regulated by pro-inflammatory mediators secreted by macrophages, including cytokines such as IL-1β, IL-6, TNF-α and the enzymes COX-2 and iNOS that produce PGE2 and NO, respectively. NF-κB is the main transcription factor involved in inflammatory processes, which translocates to the nucleus after phosphorylation of IκB, activating the expression of pro-inflammatory genes [71]. Terpenes and terpenoids exert their inflammatory actions through reducing activity of NF-κB pathway, thus ameliorating expression of pro-inflammatory mediators [77,84].

Moreover, terpenes may induce PPARγ, a transcription factor that promotes anti-inflammatory responses and resolution of inflammatory processes, by acting like PPARγ agonists, thereby inhibiting the expression of pro-inflammatory cytokines and promoting immune cells polarization toward anti-inflammatory phenotypes [84,87].

Another target that several compounds listed in the Table 2 have in common is the mitogen-activated protein kinase (MAPK) family. MAPK pathways are activated by primary inflammatory stimuli and cytokines, including TNF-α and IL-1β, through TLRs, TNF receptors (TNFR) or IL-1 receptor (TIR). The NF-κB pathway and several inflammatory mediators are regulated by MAPK cascades, which are activated by phosphorylation. JNK is involved in iNOS expression and AP-1 activation through c-Jun subunit phosphorylation; ERK promotes iNOS expression, IKK activation and production of several cytokines, including TNF-α; and p38 MAPK leads to iNOS production and the expression of pro-inflammatory mediators such as COX-2, IL-1β, TNF-α, IL-6 and, IL-8 [88,89]. Figure 5B summarizes the anti-inflammatory mechanism of action of these compounds, and Table 2 summarizes all the aforementioned anti-inflammatory compounds with their mechanisms of action.

## 4. Bioavailability

The anti-inflammatory capacity of these compounds has been mostly studied in vitro, without considering the possible effects human digestion and metabolism could have on their anti-inflammatory potential. For example, flavonoids are metabolized by conjugation reactions, such as glucuronidation, methylation or sulfation in the small intestine and liver. Flavonoids in their non-glycosylated form (aglycones) can be absorbed directly by passive diffusion in the small intestine, while their glycosylated form must be hydrolyzed by intestinal enzymes or microflora [122]. Furthermore, there are differences in the immunomodulation exerted by glycosylated forms of flavonoids and their corresponding aglycones [123]. Regarding omega 3 fatty acids, they are present in fish as free fatty acids and linked to triglycerides. Lipases hydrolyze glycerol and free the fatty acids that can be absorbed by simple diffusion in the presence of bile, and therefore the bioavailability of these fatty acids is very high [56].

## 5. An Anti-Inflammatory Dietary Pattern: What and How Much to Eat

We have described in the previous section that the most important anti-inflammatory components present in diet are ω-3 PUFA, phenolic compounds and terpenes/terpenoids. They can be found in a large variety of products which are usually consumed in MD, ND, Washoku, and other diets considered as healthy and potentially anti-inflammatory.

However, in order to obtain the benefits of these molecules, they should be consumed in high enough quantities. For instance, only high doses of EPA and DHA—at least 2 g per day—have achieved anti-inflammatory effects in humans [75]. In fact, data from a dose-response study in healthy participants with an EPA-rich supplement showed that an EPA intake of 1.35 g/day for 3 months was not sufficient to influence ex vivo PGE2 production by LPS-stimulated mononuclear cells, whereas an EPA intake of 2.7 g/day significantly decreased PGE2 production. This study suggests a threshold for an anti-inflammatory effect of EPA of somewhere between 1.35 and 2.7 g EPA per day [75]. Furthermore, combined intakes of EPA and DHA in the diet are preferred because this leads to an enrichment in EPA and DHA content in the cell membranes in a time- and dose-dependent manner and a decrease in AA content. In this sense and taking as reference an average content of 2300 mg of ω-3 PUFA (considering the sum of DHA, EPA, DPA ω-3 and Linolenic acid) present in 100 g of mackerel, a fish frequently eaten in the Mediterranean area and Nordic regions, a portion of 150 g/day would cover the quantity of the ω-3 fatty acids needed to exert their anti-inflammatory properties. Furthermore, mackerel provides a ratio of ω-3 PUFA/n-6 PUFA of 6.1. This portion size of 150 g of salmon would also cover that quantity of ω-3 PUFA maintaining a good ω-3 PUFA/ω-6 PUFA ratio of 5.4.

Cacao is a major source of flavonoids. Dark chocolate (cacao% > 70%) provides a large quantity of cacao and is therefore a good source of these phenolic compound, particularly of ferulic acid, quercetin, epicatechin and catechin, and resveratrol. However, the quantities of each of these molecules necessary to act as anti-inflammatory players individually would imply the intake of huge quantities of dark chocolate. For instance, ferulic acid has demonstrated a statistically significant reduction in the inflammatory markers hs-CRP and TNF-α by using a dose of 1000 mg/day, which would mean eating six bars of dark chocolate daily. However, since cocoa and dark chocolate exhibit a combination of different phenolic compounds, several clinical trials with daily intakes between 30–50 g of dark chocolate (1/3 bar) has shown reduction in hs-CRP, TNF-α and IL-6 and an increase in the anti-inflammatory marker IL-10 [124,125].

Because obtaining the anti-inflammatory quantity of each compound through the intake of only one food is not possible, the most reasonable strategy is combining the intake of several foods rich in them. For instance, the combination of different vegetables such as aubergine, artichoke and celery spiced up with olive oil, vinegar, rosemary, oregano or cumin provides ferulic acid, kaempferol, caffeic acid, apigenin, gallic acid and carnosol. Keeping this in mind, we have proposed an example of a one-day anti-inflammatory diet based on five meals (Table 3). The total content of anti-inflammatory compounds provided by this diet is outlined in Table 4.

It is important to bear in mind that the diet must be adapted to the tastes of each individual because the enjoyment of meals is key. On the other hand, when we eat foods with a certain pro-inflammatory activity, such as red meat or products rich in animal fat, it is important to combine them with foods that can counteract this damage, such as salads and fruits or by drinking tea or red wine, that contain anti-inflammatory compounds.

## 6. Inflammation, Oxidative Stress, and Hormesis

Chronic pathologies are normally presented by inflammation and oxidative stress due to a dysregulation caused by an increase of free radicals and pro-inflammatory factors. Their relationship is thought to begin with an increase in oxidative stress that ends up stimulating the activation of inflammatory factors which, when chronic, affect the progress of the pathology even more negatively (Oxidative Stress, Inflammation, and Disease Shampa Chatterjee). In this review, we have discussed the ability of different compounds present in diet to reduce inflammatory processes, not only by reducing the presence of pro-inflammatory products but also by promoting the presence of anti-inflammatory factors.

Some authors such as Edward J. Calabrese et al. postulated the hormetic effect of chemical compounds more than 30 years ago. They observed how small damaging stimuli at the cellular and tissue level triggered a protective response that positively affected DNA repair, life span or tumor incidence (the occurrence of chemically induced hormesis). The hormetic effect has gained interest over the years, and it is being postulated that the possible mechanism of action of some polyphenols against neurodegenerative pathologies such as Parkinson’s or ALS is due to this phenomenon (Calabrese et al., 2010, Antiox Redox signal 13,1763;) (Siracusa et al., Antioxidants 2020, 9 (9): E824) (Demonstrated hormetic mechanisms supposedly serve the effects induced by riluzole in neuroprotection against amyotrophic lateral sclerosis (ALS): Implications for research and clinical practice). In fact, it appears that stimuli that produce mild inflammation triggers an increase in the production of anti-inflammatory compounds (Aging and Parkinson’s disease: inflammation, neuroinflammation and biological remodeling as key factors in the pathogenesis).

Eating lots of fruits and vegetables can improve health, and many people will point out the antioxidants in these foods. This reasoning is logical because it seems that diseases such as cancer, cardiovascular disease or diabetes involve cell damage caused by free radicals that antioxidants neutralize. However, the story of antioxidants is not that simple. In fact, there are controlled trials in animals and humans with antioxidant vitamins C, E, and A that have failed to prevent or ameliorate disease. Thus, how do fruits and vegetables support health? Plants have developed strategies to protect themselves over millions of years and also to promote the health of who or what propagates them. The bitter-tasting chemicals made by plants act as natural pesticides. When we eat plant-based foods, we consume low levels of these toxic chemicals, which causes low levels of stress in the body’s cells, in the same way that exercise or lack of food over long periods of time do. The cells do not die; in fact, they become stronger because their stress response strengthens their ability to adapt to increased levels of stress. This process is called hormesis and could be the possible mechanism of action of why vegetables exert beneficial effects on our health (Sci Am. 2015 July; 313 (1): 40–45.).

## 7. Conclusions

Inflammation is a key physiological process in immunity and tissue repair. However, during aging it becomes increasingly more chronic. In addition, we found that certain foods such as saturated fats have pro-inflammatory activity. Taking this into account, in this review we have proposed some dietary guidelines as well as a list of compounds present in foods with anti-inflammatory activity. It must be taken into account that the amounts used in the studies that detect anti-inflammatory activity of these compounds are very high, and the intake of a single food to achieve its anti-inflammatory power is not feasible. However, the combination of foods rich in compounds with anti-inflammatory activity could exert beneficial effects during aging and in pathologies associated with inflammation and in reducing the detrimental effects of foods with pro-inflammatory activity. Therefore, we can conclude that the compounds in our diet with anti-inflammatory activity could help alleviate the inflammatory processes derived from diseases and unhealthy diets, and thereby promote healthy aging. Thus, we can use diet not only for nourishment, but also as medicine.

## Figures and Tables

**Figure 1 biomedicines-09-00922-f001:**
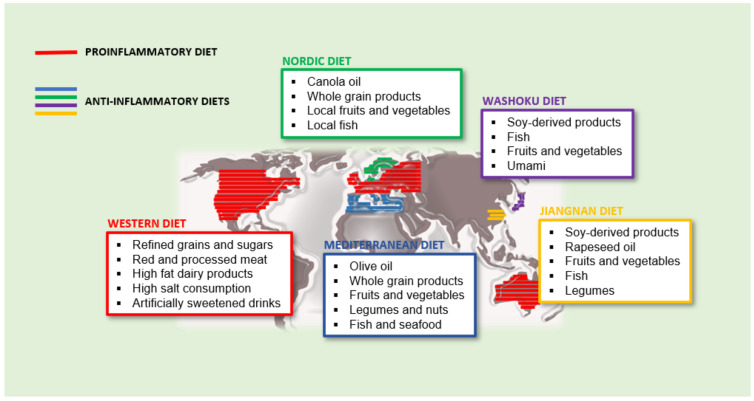
Proinflammatory and anti-inflammatory diets and its main distribution in the world.

**Figure 2 biomedicines-09-00922-f002:**
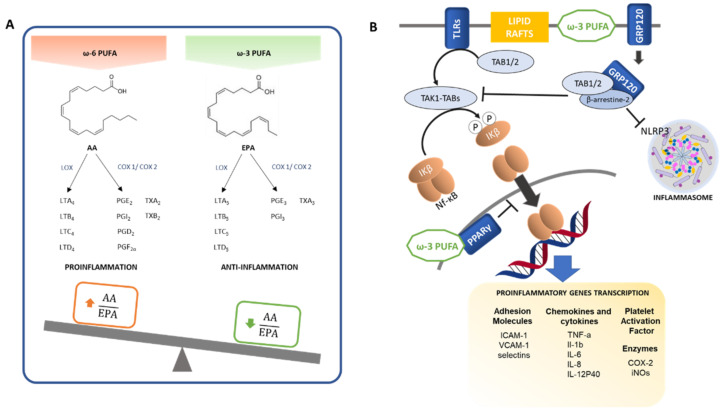
Anti-inflammatory actions of ω-3 PUFA. (**A**) ω-6 and ω-3 intake balance. ω-6 PUFAs are precursors of pro-inflammatory mediators while ω-3 PUFAs are precursors of anti-inflammatory mediators. (**B**) Mechanisms involved in the anti-inflammatory effects of ω-3 PUFAs. Thin arrows and bar-headed lines mean activation and inhibition of the pathway respectively. Thick arrows mean translocation.

**Figure 3 biomedicines-09-00922-f003:**
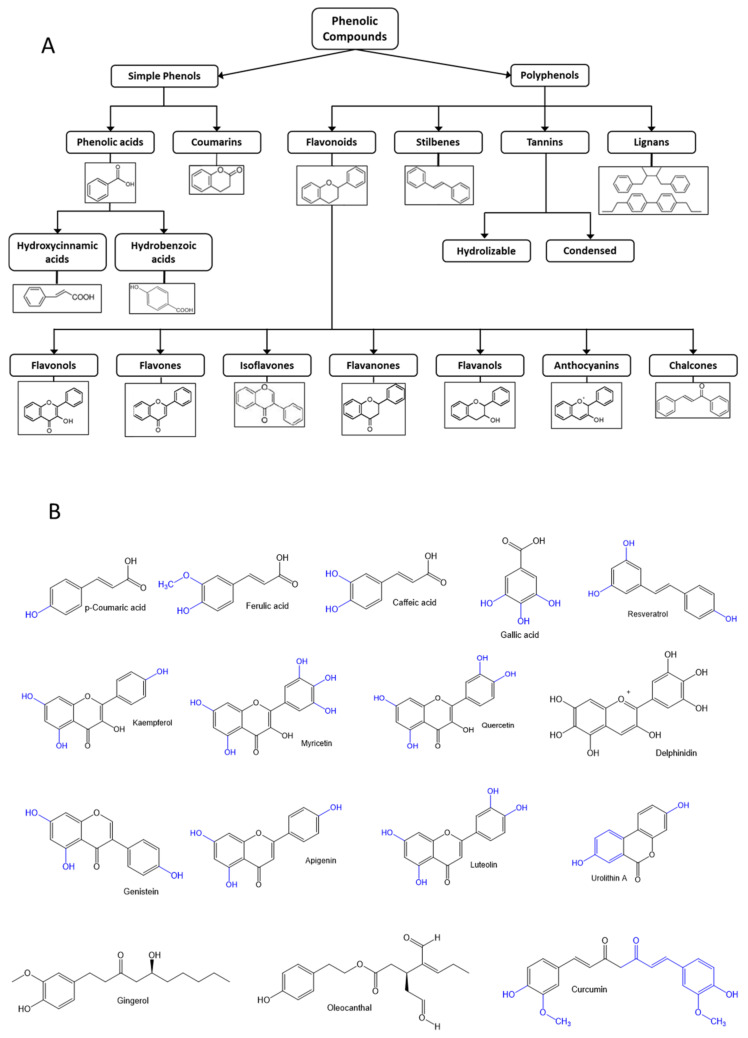
(**A**) Classification of phenolic compounds. (**B**) Chemical structure of the polyphenols found in the components of anti-inflammatory diets.

**Figure 4 biomedicines-09-00922-f004:**
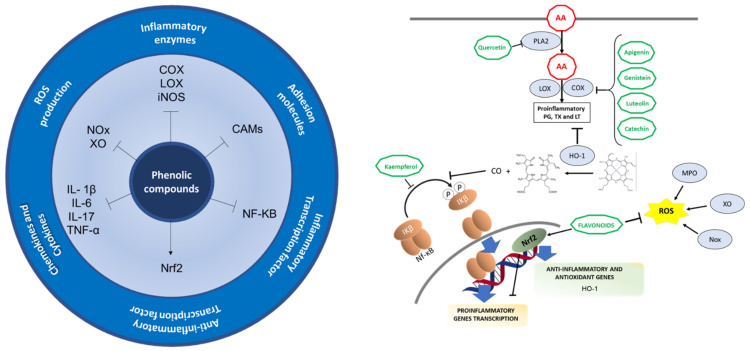
Anti-inflammatory mechanisms in which natural phenolic compounds are involved. Arrows and bar-headed lines mean activation and inhibition of the pathway respectively.

**Figure 5 biomedicines-09-00922-f005:**
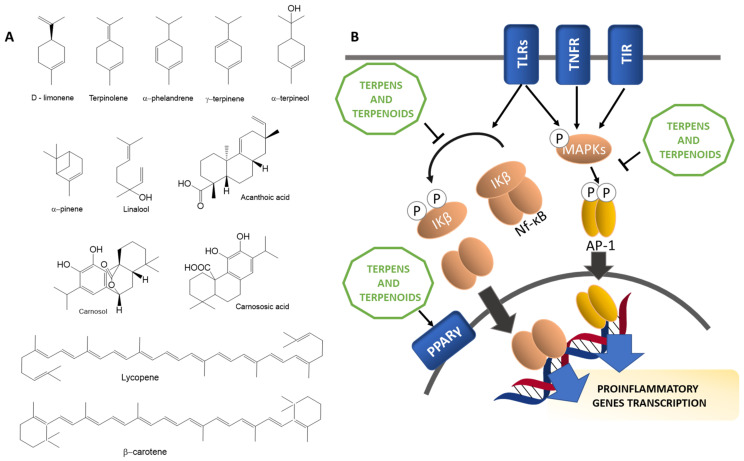
(**A**) Chemical structure of the main terpenes and terpenoids present in anti-inflammatory diets. (**B**) Anti-inflammatory mechanisms in which terpenes and terpenoids are involved. Thin arrows and bar-headed lines mean activation and inhibition of the pathway respectively. Thick arrows mean translocation.

**Table 1 biomedicines-09-00922-t001:** Representation of the aliments present in the aforementioned diets with their main anti-inflammatory compounds.

Polyphenols	Ref	Source (mg/100 g Fresh or 100 mL)
Ferulic acid	[53]	Green raw Olive (3), Aubergine (3), Refined Flour (14), Whole Grain Flour (72), Dark Chocolate (24)
Quercetin	[53]	Dark Chocolate (25), Blank Elderberry (42), Capers (33), Raw Shallot (2)
Kaempferol	[53]	Capers (104), Cumin (39), Common bean [Black] (2), Common bean [White] (0,9) and (o-glucoside 40)
Cyanidin	[53]	Common bean [Black] (1.6), Strawberry (0.5), Red raspberry (0.5), Black Olive (11), Red lettuce (0.6), Black elderberry (o-glucoside 794)
Epicatechin	[53]	Cocoa powder (158), Cider apple (29), Black Olive (4.1), broad bean (23), Blackberry (12)
Caffeic acid	[53]	Bread Rye and Whole Grain Flour (0.8), Parboiled Rice (0.34), Black Chokeberry (141), Dried Common Sage (26), Oregano (21), Rosemary (10), Green Chicory (2.3), Dried Common Thyme (21), Raw Potato (1,6)
Resveratrol	[53]	Lingonberry (3), Red currant (1.6), Red Wine (3–0.25), Red Wine (o.gluciside 0.3), Green Grape (0.24), Dark Chocolate (o-glucoside 0.1)
Tangeretin	[53]	Orange Juice (0,03–0.06)
Catechin	[53]	Red wine (6.8), Dark Chocolate (21), Powder Cocoa (108), Barley (1.2), Cider apple (5.6), Rhubarb (2.2), Plum (4.6), Rosemary (2.7), Pecan Nut (7,2), Broad been (13), Lentils (0.3)
Urolithin	[53]	Pomegranate Juice (Punicalagin 44), Red Raspberry (76)
Apigenin	[53]	Extra Virgin Olive Oil (1.2), Commo Sage (2.4), Oregano (3.3), Rosemary (0.6), Black Olive (o-glucoside 8.2), Artichoke (o-glucoside 7.4), Celery Seed (o-glucoside 111), Lentils (o-glucoside 0.62)
Carnosol	[53]	Rosemary (53)
Luteolin	[53]	Artichoke heads (42), thyme (40), olive oil extra (0.4)
Genistein	[53]	Soy (10–0.1), Black bean (0.6), White Bean (0.3)
Phlorotannins	[54]	*Fucus vesiculosus*, dry (24), *Ascophyllum nodosum* dry (16)
Epicatechin	[53]	Cocoa Powder (158), Apple Cider (29), Black Olive (4.1), Broad Bean (23), Blackberry (12)
Gallic acid	[53]	Blackberry (4.7), Red Wine (3.6), Black Tea (4.7), Cloves (458), Oregano (5.2), Vinegar (2.6), Chestnut (480), Chicory (26)
Gingerol	[53]	Ginger Dried (187)
β-carotene	[55]	Baked Sweet Potatoes (11), Carrot Raw (8.3), Spinach (6.3), Romaine Lettuce (5.2)
ω-3 fatty acids	[56]	Cod (Gadus morhua) (40), Perca (perca fluviatis) (35), Salmon (2000), Mackerel (2300), Herring (1500)
Oleocanthal	[57]	Virgin Olive Oil (5.4)

**Table 2 biomedicines-09-00922-t002:** Outline of the forementioned anti-inflammatory compounds, their molecular mechanisms, and dose exhibition effects.

Type	Compound	Mechanism	Model	Dose Exhibiting Effects	Reference
**Polyunsaturated Fatty Acids**					
ω-3 LC-PUFA	Docosahexaenoic acid (DHA)	Reduction of intracellular ROS levels Decreased expressions of IL-6, IFN-γ, MCP-1, TNF-α, IL-1β and IL-10	PMA-stimulated monocyte cell line THP-1	300 µM	[90]
		Reduction in phosphorylation and activation of TAK1 and NF-κB Decreased TNF-α production Activation of GP120, thus blocking IKK-β/NF-κB pathway by retaining TAB1 through β-arrestin-2	TNF-α-stimulated rHypoE-7 cell line from the embryonic rat hypothalamus	100 µM	[91]
ω-3 LC-PUFA	Docosatrienoic acid (DTA)	Reduction of intracellular ROS levels Decreased expressions of IL-6, IFN-γ, MCP-1, TNF-α, IL-1β and IL-10	PMA-stimulated monocyte cell line THP-1	300 µM	[90]
ω-6 LC-PUFA	Docosadienoic acid (DDA)	Reduction of intracellular ROS levels Decreased expressions of IL-6, IFN-γ, MCP-1, TNF-α, IL-1β and IL-10	PMA-stimulated monocyte cell line THP-1	300 µM	[90]
**Simple Phenolic Compounds**					
Hydrobenzoic acid	Urolithin A	Downregulation of COX-2 and iNOS expression and decreased PGE_2_ an NO generation Inhibition of TNF-α and IL-6 production Prevention of IκB-α phosphorylation and p65 translocation into the nucleus Decreased PI3K and Akt phosphorylation, thus amelioration of PI3K/Akt/NF-κB pathway activation	Primary human osteoarthritis chondrocytes	30 µM	[92]
		Inhibition of Akt and JNK phosphorylation Suppression of NF-κB and AP-1 activation Inhibition of NADPH oxidase (NOX) and reduction of intracellular accumulation of ROS Reduction in TNF-α and IL-6 production	LPS-activated RAW264.7 macrophages ICR-mouse resident peritoneal macrophages	10–20–40 µM	[93]
		Inhibition of phosphorylation of MAPK pathway members (ERK1/2, JNK, p38) Suppression of COX-2, iNOS and MMPsInhibition of p65 phosphorylation and NF-κB pathway activation	IL-1β stimulated primary chondrocytes from Sprague Dawley rats	15 µM	[94]
Hydrobenzoic acid	Gallic acid	Inhibition of xanthine oxidase Inhibition of COX-2 Inhibition of TNF-α, IL-6	LPS-stimulated THP-1 cells	10 µM	[95]
Hydroxycinnamic acid	*p*-Coumaric acid	Suppression of COX-2, iNOS, TNF-α and IL-1β expression Inhibition of IκB-α phosphorylation and nuclear translocation of p65, preventing activation of NF-κB pathway Inhibition of phosphorylation of ERK1/2 and JNK	LPS-activated RAW264.7 macrophages	50–100 µg/mL	[96]
Hydroxycinnamic acid	Ferulic acid	Inhibition of xanthine oxidase Inhibition of COX-2 Inhibition of TNF-α, IL-6	LPS-stimulated THP-1	10 µM	[95]
		Inhibition of NF-κB translocation into the nucleus Inhibition of IKK/TAK1 activation Inhibition of Nrf2 pathway by impairing the reduction of cysteine residues in Keap1 Reduced MnSOD and TLR4 protein levels	LPS-activated RAW264.7 macrophages	100 µM	[97]
Hydroxycinnamic acid	Caffeic acid	Inhibition of xanthine oxidase Inhibition of COX-2 Inhibition of TNF-α, IL-6	LPS-stimulated THP-1	10 µM	[95]
Hydroxicinnamic acid	Curcumin	Inhibition of the synthesis of proinflammatory mediators IL-1β, IL-6, and iNOS and NO production Increased production of anti-inflammatory cytokines IL-6, IL-10, and Arg-1 Microglial polarization to M2 phenotype (decreased iNOS and increased CD206 immunoreactivity) Increased TREM2 expression and inhibition of TLR4 expression and p65 NF-κB phosphorylation	Murine BV2 microglial cell line	5–10 µM	[98]
**Polyphenols**					
**Stilbene**	**Resveratrol**	Inhibition of NF-κB and AP-1 activation Downregulation of ICAM-1, VCAM-1, E-selectin expression Downregulation of MCP-1, M-CSF expression	Human umbilical vein endothelial cells (HUVEC)	10–25 µM	[99]
		Inhibition of IL-1β-induced VEGF, COX-2, MMP-3, MMP-9 gene expressionPrevention of IκB-α degradation (by inhibiting proteasome function) and NF-κB activation	Isolated primary human articular chondrocytes	100 µM	[100]
		Inhibition of ROS (O_2_^−^, H_2_O_2_) production Inhibition of arachidonic acid release, COX-2 induction and PGE_2_ synthesis	CD-1-mouse resident peritoneal macrophages stimulated by LPS or PMA	30 µM	[101]
**Flavonol**	**Kaempferol**	Suppression of COX-2 and iNOS expression Inhibition of PGE_2_ production	LPS-activated RAW264.7 macrophages	15–25 µM	[102]
		Inhibition of NF-κB and AP-1 activation Downregulation of ICAM-1, VCAM-1, E-selectin expression Inhibition of MCP-1 synthesis and secretion	Human umbilical vein endothelial cells (HUVEC)	10–25 µM	[99]
		Inhibition of PGE_2_ production	LPS-stimulated human mononuclear cells	10–100 µM	[103]
**Flavonol**	**Myricetin**	Reduction in NF-κB and AP-1 activation Downregulation of ICAM-1, VCAM-1, E-selectin expression Inhibition of MCP-1 synthesis and secretion	Human umbilical vein endothelial cells (HUVEC)	10–25 µM	[99]
**Flavonol**	**Quercetin**	Inhibition of NO production and iNOS protein expression Stabilization of IκB-α and IκB-β preventing NF-κB activation Inhibition of activation of MAPK (Erk1/2, p38, JNK/SAPK) Inhibition of phosphorylation and activation of JAK/STAT	LPS-activated RAW264.7 macrophages	50 µM	[104]
		Inhibition of NF-κB and AP-1 activation Downregulation of ICAM-1, VCAM-1, E-selectin expression Downregulation of MCP-1, M-CSF expression	Human umbilical vein endothelial cells (HUVEC)	10–25 µM	[99]
		Inhibition of iNOS expression and NO production Attenuation of IκB-α phosphorylation and NF-κB activation Inhibition of DNA binding activity of AP-1 and STAT1 Induction of heme oxygenase-1 (HO-1) expression, possibly through Src family members	Murine BV2 microglial cell line	3–10–30 µM	[105]
Flavanol	Epigallocatechin 3-gallate/epigallocatechin gallate (EGCG)	Suppression of TNF-α, IL-1β, and IL-6 expression	Human dermal fibroblasts	50 µM	[106]
		Downregulation of COX-2 and iNOS expression	LPS-activated RAW264.7 macrophages	20–40 µM	[107]
Flavanol	(-)-epicatechins	Inhibition of the production of NO, PGE_2_, TNF-α, IL-6	LPS-activated RAW264.7 macrophages	5–25–50 µM	[108]
Flavanol	Catechin	Inhibition of gene expression of pro-inflammatory cytokines (IL-1α, IL-1β, IL-6, IL-12p35) and enzymes (COX-2, iNOS) Up-regulation of anti-inflammatory cytokines (IL-4, IL10) Inhibition of the activation of NF-κB, AMPK, FOXO3a and SIRT1	TNF-α induced 3T3-L1 adipocytes	10–25–50–100 μg/mL	[109]
Flavone	Apigenin	Suppression of COX-2 and iNOS expression Inhibition of PGE_2_ and NO production Inhibition of IKK activity preventing NF-κB activation	LPS-activated RAW264.7 macrophages	5–15–25 µM	[102]
Flavone	Luteolin	Suppression of TNF-α, IL-6, iNOS, COX-2 gene expression Decreased Akt and IKK phosphorylation Blockade of NF-κB and AP-1 activation Inhibition of ROS generation	LPS-activated RAW264.7 macrophages	5–10–25 µM	[110]
Isoflavone	Genistein	Suppression of COX-2 and iNOS expression Inhibition of PGE_2_ production	LPS-activated RAW264.7 macrophages	15–25 µM	[102]
		Suppression of protein expression of iNOS, COX-2, TNF-α, IL-1β, IL-6 Prevention of MAPKs and NF-κB pathways activation Up-regulation of G protein-coupled estrogen receptor (GPER) gene expression	Murine BV2 microglial cell line Primary microglia cell culture	10 µM	[111]
Anthocyanins (Flavonoid)	Anthocyanins-rich extract from berries	Reduction in TNF-α secretion Inhibition of NF-κB translocation into the nucleus Reduction of cellular ROS levels Downregulation of IL-1β and NADPH oxidases 1 (NOX-1) expression	LPS-activated RAW264.7 macrophages	20 µg/mL	[112]
Anthocyanins (Flavonoid)	Anthocyanins-rich extract	Inhibition of COX-2 and iNOS expression, and inhibition of PGE_2_ and NO production Inhibition of TNF-α and IL-1β expression Prevention of IκB-α degradation and NF-κB translocation Inhibition of Akt and MAPKs (ERK, JNK, p38) phosphorylation	Murine BV2 microglial cell line	50–100 µg/mL	[113]
Anthocyanins (Flavonoid)	Delphinidin	Inhibition of COX-2 expression Inhibition of phosphorylation of c-Jun, thus inhibiting AP-1 Inhibition of NF-κB activation by blocking IκB-α degradation and p65 translocation Inhibition of the three MAPK signalling pathways (JNK, ERK, p38)	LPS-activated RAW264.7 macrophages	50–75–100 µM	[114]
Other compunds	Gingerol	Decreased PGE_2_ and NO secretion Decreased COX-2, iNOS, TNF-α, IL-1β, IL-6 expression Inhibition of NF-κB activation by suppressing phosphorylation of IκBα and p65	LPS-activated RAW264.7 macrophages	50–100–200–300 μg/mL	[115]
		Decreased of ROS production (in PMNs) Inhibition of NO production (in macrophages) Inhibition of PGE_2_ production (in macrophages)	Human polymorphonuclear neutrophils (PMN) LPS-activated RAW264.7 macrophages	6 µM	[116]
Other compunds	Oleocanthal	Inhibition of iNOS expression and NO production Suppression of MIP-1α, IL-6, IL-1β, TNF-α,and GM-CSF expression	Murine macrophages J774 Murine chondrocytes ATDC5	15–50 µM	[117]
**Terpenes and Terpenoids**					
Terpenoid (carotenoid)	Lycopene	Downregulation of TNF-α, IL-1β, IL-6, iNOS, and COX-2 expression Inhibition of PGE_2_ and NO production Decreased JNK, ERK and NF-κB protein expression	LPS-stimulated SW 480 human colorectal cancer cells	20–30 µM	[118]
Terpenoid (carotenoid)	β-carotene	Suppression of expression of COX-2, iNOS and TNF-α and IL-1β pro-forms PGE_2_, NO and ROS production Inhibition of IκB-α degradation and NF-κB activation	LPS-activated RAW264.7 macrophages BALB/c mice peritoneal macrophages stimulated by LPS	50 µM	[119]
Diterpene	Carnosol and carnosic acid	Inhibition of LOX and mPGES-1 (microsomal PGE_2_ synthase-1) enzymes and formation of arachidonic acid-derived eicosanoids	Activated human primary monocytes and neutrophils	3–30 µM	[120]
		Reduction in NO and PGE_2_ production Inhibition of COX-2 activity Downregulation of iNOS, IL-1α, IL-6 or CXCL10/IP-10 genes expression (in macrophages) Downregulation of CCL5/RANTES and CXCL10/IP-10 gene expression (in SW1353 cells) Downregulation of interleukine (IL-1α, IL-1β, IL-6) and chemokine genes expression (CXCL8/IL-8, CCL20/MIP-3α, CCL5/RANTES) Inhibition of NF-κB (p65) translocation into the nucleus	LPS-stimulated RAW264.7 macrophages IL-1β activated chondrosarcoma SW1353 cells IL-1β activated primary human articular chondrocytes	12.5 µM	[121]

**Table 3 biomedicines-09-00922-t003:** Example of a one-day meal plan with high anti-inflammatory effects.

**Breakfast**		**g**	**Anti-Inflammatory Compound (mg/100 g)**		**mg**
**Wholemeal bread**		60	72	Ferulic Acid	43.20
Olive oil		10	5.4	Oleocanthal	0.54
			1.2	Apigenin	0.12
			0.4	Luteolin	0.04
Soy milk		200	3.7	Genistein	7.40
Blackberries		50	4.7	Gallic Acid	2.35
			12	Epicathechin	6.00
Red currant		50	1.6	Resveratrol	0.80
**Morning Snack**		**g**	**Anti-inflammatory compound (mg/100 g)**		**mg**
**Pommegranate juice**		200	44	Punicalgin	88.00
Pecan nuts		30	7.2	Catechin	2.16
Chesnuts		40	480	Gallic Acid	192.00
**Lunch**		**g**	**Anti-inflammatory compound (mg/100 g)**		**mg**
**Salad**	**Rhubarb**	122	2.2	Catechin	2.68
	Black olives	16	4.1	Epicatechin	0.65
			8.2	Apigenin	1.31
	Green olives	16	3	Ferulic Acid	0.48
	Capers	8.6	33	Quercetin	2.83
			104	Kaempferol	8.94
	Tomate	100	4.2	Lycopene	4.20
	Vinegar	10	2.6	Gallic Acid	0.26
	Olive oil	10	5.4	Oleocanthal	0.54
			1.2	Apigenin	0.12
			0.4	Luteolin	0.04
Lentils and vegetables stew	Lentils	40	0.3	Catechin	0.12
	Black bean	20	0.6	Genistein	0.12
			2	Kaempferol	0.40
	Carrots	50	8.3	Beta-carotene	4.15
	Tomate	90	4.2	Lycopene	3.78
	Shallot	25	2	Quercetin	0.50
	Olive oil	10	5.4	Oleocanthal	0.54
			1.2	Apigenin	0.12
			0.4	Luteolin	0.04
	Oregano	1	21	Caffeic Acid	0.21
			3.3	Apigenin	0.03
			5.2	Gallic Acid	0.05
	Cumin	1	39	Kaempferol	0.39
	Cloves	1	458	Gallic Acid	4.58
Rye Bread		60	0.8	Epicathechin	0.48
Red wine		100	3	Resveratrol	3.00
			6.8	Catechin	6.80
			3.6	Gallic Acid	3.60
Plum		200	4.6	Catechin	9.20
Dark chocolate		20	25	Quercetin	5.00
			0.1	Resveratrol	0.02
			21	Catechin	4.20
			24	Ferulic Acid	4.80
**Afternoon Snack**		**g**	**Anti-inflammatory compound (mg/100 g)**		**mg**
**Orange juice**		200	0.06	Tangerin	0.12
Pecan nut		30	7.2	Catechin	2.16
**Dinner**		**g**	**Anti-inflammatory compound (mg/100 g)**		**mg**
**Grilled salmon with vegetables**	**Salmon**	150	2000	ω-3 PUFA	3000.00
	Olive oil	10	5.4	Oleocanthal	0.54
			1.2	Apigenin	0.12
			0.4	Luteolin	0.04
	Aubergine	150	3	Ferulic Acid	4.50
	Artichocke	70	7.4	Apigenin	5.18
	Rosemary	1	10	Caffeic Acid	0.10
			2.7	Catechin	0.027
			0.6	Apigenin	0.006
Bread rye		60	0.8	Epicathechin	0.48
Red wine		100	3	Resveratrol	3.00
			6.8	Catechin	6.80
			3.6	Gallic Acid	3.60
Green Grape		200	0.4	Resveratrol	0.80
Dark chocolate		20	25	Quercetin	5.00
			0.1	Resveratrol	0.02
			21	Catechin	4.20
			24	Ferulic Acid	4.80

**Table 4 biomedicines-09-00922-t004:** Anti-inflammatory compounds present in the outlined meal plan in Table 3.

Compound	Total mg
Ferulic Acid	57.78
Oleocanthal	2.16
Apigenin	7.00
Genistein	7.52
Gallic Acid	204.09
Epicathechin	7.62
Resveratrol	7.64
Punicalgin	88.00
Catechin	41.02
Quercetin	13.30
Kaempferol	9.73
Lycopene	7.98
Luteolin	0.16
beta-carotene	4.15
Caffeic Acid	0.31
Tangerin	0.12
n-PUFA	3000.00

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
