# Peer review of "Anti-Inflammatory Properties of Diet: Role in Healthy Aging"

_biomedicines, 2021, doi:10.3390/biomedicines9080922_

Round 1

Reviewer 1 Report

It is an important topic for health promotion. However, there are some comments that I would like to give.

  1. This article mainly summarizes the anti-inflammatory effects of indicated diet. However, there are limited direct evidence about the effects on aging.
  2. The proposed healthy diet by authors might need scientific validation.
  3. The contents of table 1 need reorganization for readability.
  4. Some words are suggested corrections. Line 178 “osteoporosis” (osteoarthritis); Line 261 “VFA” (visceral fat area (VFA)); Line 481-482 “Figure 4B” (Figure 5); Line 483 “wit teir” (with there?); Line 486 “Figure4” (Figure 5); Line 490 “Biodisponibilidad” (Bioavailability); Line 503 “Acabar con una conclusión” (delete???); Line 558 “Figure 3” (Table 3).

Author Response

Reviewer 1

It is an important topic for health promotion. However, there are some comments that I would like to give.

  1. This article mainly summarizes the anti-inflammatory effects of indicated diet. However, there are limited direct evidence about the effects on aging.

Thank you for your comment. Although we have not directly linked aging to each diet, in the introduction we outlined the hallmarks of aging and the role of inflammation in aging, which are linked to the compounds of diet in each paragraph. However, taking into account the reviewer’s comment, we have modified the title to better suit the content of the article.

  1. The proposed healthy diet by authors might need scientific validation.

The diet proposed in the review was created by one of the authors, who is a nutritionist specializing in the effects of polyphenolic compounds on health. It is only an example of a possible diet rich in anti-inflammatory compounds, but if the reviewer does not find it appropriate, we will eliminate it.

  1. The contents of table 1 need reorganization for readability.

Thank you for your suggestion. The table has been modified to allow better visualization of the data.

  1. Some words are suggested corrections. Line 178 “osteoporosis” (osteoarthritis); Line 261 “VFA” (visceral fat area (VFA)); Line 481-482 “Figure 4B” (Figure 5); Line 483 “wit teir” (with there?); Line 486 “Figure4” (Figure 5); Line 490 “Biodisponibilidad” (Bioavailability); Line 503 “Acabar con una conclusión” (delete???); Line 558 “Figure 3” (Table 3).

Thank you for your observations. We have made the orthographic changed outlined.

Reviewer 2 Report

Interplay and coordination of redox interactions with endogenous and exogenous antioxidant defence systems  is an emerging area of reserach interest in anticancer and antidegenerative therapeutics. Moreover, particular attention has been given to providing an assessment of the quantitative features of the dose-response relationships and underlying mechanisms that could account for the biphasic nature of the hormetic response after exposure to redox active agents, such as free radical oxygen species and their impact in inflammatory/antinflammatory pathways. The hormetic dose response should be seen as a reliable feature of the dose response for oxygen free radicals and their redox regulated transcriptional factors  as well as  antioxidant compounds and appears to have an important impact  on brain pathophysiology and stress resistance mechanisms to oxidative and inflammatory insult and neurodegenerative damage.
This is an interesting paper.  The study is well-conceived and well-executed. This reviewer is satisfied with the significance of this study, the care in which the study was performed, and the implications of the results for human health.  However, although the results presented are convincing, the work raises some concerns which will need to be addressed. The questions posed are of extremely high interest, but the paper does not give adequate definitive information, therefore pending addressing some major question is possible to accept for publication.

Minor concerns:

1.    Preconditioning signal leading to cellular protection through Hormesis is an important redox dependent aging-associated to free radicals species accumulation, inflammatory responses involved in neurodegenerative/ neuroprotective mechanisms. This aspect should be highlighted in the discussion and references properly added (See Calabrese et al., 2010, Antiox. Redox Signal 13,1763; Siracusa et al., Antioxidants 2020, 9(9):E824; Calabrese V. et al., Free Radical Biology and Medicine 20, 391-397, 1996).

2. Given the relationship between polyphenol compounds, redox status and the vitagene network and its possible biological relevance in neuroprotection, Authors while interpetrating results should discuss appropriately this aspect and make proper connection with emerging principles of hormesis ( Calabrese et al., Ageing Res Rev. 2018, 42:40-55; Calabrese EJ e tal., Ageing Res Rev. 2021 Feb 8;67:101273; Mattson et al., Int Rev Neurobiol. 2020;155:271-301; Brunetti et al., Int J Mol Sci. 2020 Apr 8;21(7):2588). 

Author Response

Reviewer 2

Interplay and coordination of redox interactions with endogenous and exogenous antioxidant defence systems  is an emerging area of reserach interest in anticancer and antidegenerative therapeutics. Moreover, particular attention has been given to providing an assessment of the quantitative features of the dose-response relationships and underlying mechanisms that could account for the biphasic nature of the hormetic response after exposure to redox active agents, such as free radical oxygen species and their impact in inflammatory/antinflammatory pathways. The hormetic dose response should be seen as a reliable feature of the dose response for oxygen free radicals and their redox regulated transcriptional factors  as well as  antioxidant compounds and appears to have an important impact  on brain pathophysiology and stress resistance mechanisms to oxidative and inflammatory insult and neurodegenerative damage.      
This is an interesting paper.  The study is well-conceived and well-executed. This reviewer is satisfied with the significance of this study, the care in which the study was performed, and the implications of the results for human health.  However, although the results presented are convincing, the work raises some concerns which will need to be addressed. The questions posed are of extremely high interest, but the paper does not give adequate definitive information, therefore pending addressing some major question is possible to accept for publication.

Minor concerns:

  1. Preconditioning signal leading to cellular protection through Hormesis is an important redox dependent aging-associated to free radicals species accumulation, inflammatory responses involved in neurodegenerative/ neuroprotective mechanisms. This aspect should be highlighted in the discussion and references properly added (See Calabrese et al., 2010, Antiox. Redox Signal 13,1763; Siracusa et al., Antioxidants 2020, 9(9):E824; Calabrese V. et al., Free Radical Biology and Medicine 20, 391-397, 1996).

Thank you for your kind words and suggestions. We have added a section on inflammation, oxidative stress, and hormesis using the articles suggested by the reviewer.

  1. Given the relationship between polyphenol compounds, redox status and the vitagene network and its possible biological relevance in neuroprotection, Authors while interpetrating results should discuss appropriately this aspect and make proper connection with emerging principles of hormesis ( Calabrese et al., Ageing Res Rev. 2018, 42:40-55; Calabrese EJ e tal., Ageing Res Rev. 2021 Feb 8;67:101273; Mattson et al., Int Rev Neurobiol. 2020;155:271-301; Brunetti et al., Int J Mol Sci. 2020 Apr 8;21(7):2588). 

In the section of hormesis we have related the anti-inflammatory properties of polyphenols to hormesis, attributed their protective properties to this phenomenon.

We were however not able to modify the references and have therefore highlighted the newly added citations in yellow in the text and ask kindly that the journal add them for us. Thank you.

Round 2

Reviewer 1 Report

The manuscript was mostly corrected according to suggestions.